# Bioavailability of Iron and the Influence of Vitamin a in Biofortified Foods

**Paula Tavares Antunes [1], Maria das Graças Vaz-Tostes [2]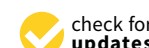, Cíntia Tomáz Sant'Ana [1],**
**Renata Araújo de Faria [3], Renata Celi Lopes Toledo [4] and Neuza Maria Brunoro Costa [2,\***

[1] Food Science MSc Program, Federal University of Espirito Santo, Alegre, ES 29500-000, Brazil; paulinhat.antunes@hotmail.com (P.T.A.); cintia_santana28@hotmail.com (C.T.S.)
[2] Department of Pharmacy and Nutrition, Federal Universityof Espirito Santo, Alegre, ES 29500-000, Brazil; mgvaztostes@gmail.com
[3] University Graduate, Federal University of Espirito Santo, Alegre, ES 29 500-000, Brazil; renata.araujof@live.com
[4] Department of Nutrition and Health, Federal University of Viçosa, Viçosa, MG 36570-000, Brazil; renatacelly@yahoo.com.br
\* Correspondence: neuzambc@gmail.com

**Abstract:** Inadequate eating habits, among other factors, lead to nutritional deficiencies worldwide. Attempts have been made to control micronutrient deficits, such as biofortification of usually consumed crops, but the interaction between food components may affect the bioavailability of the nutrients. Thus, this study aimed to evaluate the effect of pro-vitamin A on the bioavailability of iron in biofortified cowpea and cassava mixture, compared to their conventional counterparts. The chemical composition of the raw material was determined, and an in vivo study was performed, with Wistar rats, using the depletion-repletion method. Gene expression of iron-metabolism proteins was evaluated. Results were compared by analysis of variance (ANOVA), followed by the Tukey test ($p < 0.05$). Biofortified cowpea (BRS *Aracê*) showed an increase of approximately 19.5% in iron content compared to conventional (BRS *Nova era*). No difference in Hemoglobin gain was observed between groups. However, the animals fed biofortified cowpea were similar to ferrous sulfate (Control group) regarding the expression of the hephaestin and ferroportin proteins, suggesting a greater efficiency in the intestinal absorption of iron. Thus, this study points out a higher efficiency of the biofortified cowpea in the bioavailability of iron, regardless of the presence of pro-vitamin A.

**Keywords:** Cowpea; cassava; bioavailability

## 1. Introduction

Developing countries are undergoing an epidemiologic transition, characterized by the growing number of obesity and non-communicable diseases and a considerable reduction of malnutrition. Paradoxically, the population still suffers from micronutrient deficiency, known as hidden hunger. Poor diet leads to the deficiency of several minerals essential for human health, highlighting iron (Fe) and zinc (Zn), as well as vitamin A, which together or alone constitute public health problems [1].

Iron deficiency is considered to be one of the most important nutritional problems in Brazil, resulting in iron deficiency anemia, and should be investigated early, since the deficiency occurs gradually and progressively in the body. One of the main symptoms is the delay of neurophysiological development, reduction of intellectual capacity, and physical weakness [2,3]. Additionally, the subclinical and clinical deficiency of vitamin A is the most prevalent vitamin deficiency, especially in children, resulting in night blindness. It has its role in cell proliferation and differentiation, and a positive effect on iron uptake in the body [4].

Strategies to combat nutritional deficiencies have been implemented along with food re-education, one of them—the biofortification of foods—developed with the intention of improving the nutritional quality of crop products, increasing the concentration of nutrients through breeding of genetically modified plants [5].

Biofortification programs have been developed worldwide, and in Brazil, the BioFort network has concentrated the efforts on the enrichment of cassava, rice, beans, cowpea, potatoes, sweet potatoes, pumpkins, corn, and wheat [6–8]. The cowpea is enriched with Fe and Zn [3,9], and cassava with pro-vitamin A, $\beta$-carotene [10].

Several factors influence the bioavailability, and therefore, it has been well studied to understand the metabolism, identifying factors that may have positive or negative actions in nutrients absorption [11,12]. Studies indicate that there is a positive interaction of vitamin A with the iron metabolism, participating in the regulation of some proteins involved in the absorption of this mineral [13,14]. The mechanisms of iron homeostasis are determined by gene expression of proteins related to iron metabolism. For instance, the duodenal cytochrome b (Dcytb) acts by reducing ferric iron ($Fe^{3+}$) to the ferrous form ($Fe^{2+}$), which is transported through the enterocyte membrane by the divalent metal transporter 1 (DMT1). The Ferroportin exports iron to plasma cells, converting ferrous iron ($Fe^{2+}$) back to ferric iron ($Fe^{3+}$) through hephaestin to be bound to transferring [15]. Therefore, it is important to evaluate this interaction effect and the factors that positively influence it, for a better comprehensiveness of this biofortification system. Thus, this study aimed to evaluate the effect of pro-vitamin A on the bioavailability of iron in biofortified cowpea and cassava mixture, compared to their conventional counterparts, by using the depletion-repletion animal model and the gene expression of proteins related to iron metabolism.

## 2. Materials and Methods

### 2.1. Raw Material

The raw material: cowpea (*Vigna unguiculata L Walp*), cultivar BRS *Aracê*, biofortified with iron and zinc, cassava (*Manihot esculenta Crantz*), cultivar BRS *Jari*, biofortified with $\beta$-carotene (pro-vitamin A) conventional cowpea (*Vigna unguiculata L Walp*), cultivar BRS *Nova Era,* and white common rice (BRS Esmeralda) were provided by the Brazilian Agricultural Research Corporation (EMBRAPA). Conventional cassava was obtained from the local market in Alegre, ES, Brazil.

### 2.2. Preparation of Raw Materials

The rice and cowpea were cooked under pressure with deionized water (1:2 *w/v*), after being dried in an oven with air circulation for 48 h at 55 °C. The grains were crushed with a domestic blender and sieved until the consistency of flour was reached.

Cassava was boiled with deionized water for 15 min (1:2 *w/v*).

### 2.3. Determination of Food Composition

Protein concentration was determined according to Association of Official Analytical Chemists –AOAC [16]. The determination of lipids was carried out in a Soxhlet apparatus, using petroleum ether with the solvent extractor, for 4h [17]. Moisture and ashes were determined according to AOAC [17].

Soluble and insoluble fiber contents were determined by the enzymatic-gravimetric method [18]. The carbohydrate content was obtained from the difference between the total sample (100%) and the contents of protein, lipid, dietary fiber, moisture, and ash.

Total phenolic compounds content was determined by the spectrophotometric method, using the Folin–Ciocalteu reagent in an Enzyme-Linked Immunosorbent Assay (ELISA) spectrophotometer (ThermoScientific®) at 760 nm. Gallic acid was used as reference standard and the results were expressed in mg of gallic acid equivalent (mg gallic acid/100 g) [19].

The iron content was determined by nitric-perchloric acid digestion followed by atomic absorption spectrophotometry [20].

The determination of phytic acid was done according to Latta and Eskin [21]. The phytate/iron molar ratio was calculated by dividing the phytate and iron molar values of each sample and experimental diet. The molar weight of myo-inositol hexaphosphate (IP6) and iron, 660 g/mol and 55.85 g/mol respectively, were considered in the calculation.

Carotenoid extraction was performed according to Rodriguez-Amaya et al. [22], with modifications. About 5 g of samples were added to 20.0 mL of cooled acetone, homogenized in a micro-crusher (Marconi, MA 102 model, Brazil) for approximately 3 min and vacuum-filtered in a Büchner funnel, using filter paper, and the residue was maintained in the extraction tube. The extraction and filtration procedures were performed twice more on the waste until complete discoloration of the sample. Subsequently, the filtrate was transferred in three fractions to a separation funnel containing 50.0 mL of petroleum ether. After the transfer of each fraction, distilled water was added for phase separation (carotenoids in petroleum ether and acetone-water) and the bottom phase (water-acetone) was discarded. Anhydrous sodium sulfate was added to the ether extract for removal of residual water that could impair evaporation of the material. The ether extract was then concentrated using a rotary evaporator (Tecnal, TE-211 model, Brazil) at $35 \pm 1$ °C, transferred to a 25.0 mL volumetric flask, and the volume was completed with petroleum ether. Later, the extract was then stored in a hermetically sealed amber glass bottle and stored at $-18 \pm 1$ °C. For analysis, the extracts were evaporated under nitrogen gas flow and the dry residue was re-dissolved in 2.0 mL of high-performance liquid chromatography system (HPLC)-grade acetone (Tedia, Brazil). The extracts were filtered through Millex-HV filter units, in polyethylene, with 0.45 μm of porosity (Millipore, Brazil), and 50 μL were injected into the chromatographic column for analysis. Carotenoids were analyzed using a high-performance liquid chromatography system (HPLC) (Shimadzu SCL 10AT VP model), comprised of a high-pressure pump (Shimadzu LC-10AT VP), an autosampler with a loop of 50 μL (Shimadzu SIL-10AF) and a diode array detector (DAD) (Shimadzu SPD-M10A). The chromatographic conditions used were developed by Pinheiro-Sant'Ana et al. [23], and included: HPLC system, DAD; chromatographic column Phenomenex Gemini RP-18 (250 mm × 4.6 mm, 5 μm) fitted with the guard column Phenomenex (ODS, octadecyl -C18), (4 mm × 3 mm); mobile phase composed of methanol:ethyl acetate:acetonitrile (70:20:10, *v/v/v*), and flow rate of 1.7 mL/min. The chromatograms were obtained at 450 nm.

### 2.4. Preparation of Diets

The depletion diet was based on the American Institute of Nutrition -AIN-93G diet [24], by using an iron-free mineral mix. The repletion diet was similar to the depletion diet but was adjusted to provide a 12mg Fe/kg diet from ferrous sulfate ($FeSO_4$) in the control (SF) diet and from the cowpea in the test diets.

For the calculation of the test diets, the iron and protein content of the beans were used. With this, the amount of beans, cassava, and rice that were used in the diet was established, as well as the amount of albumin to be supplemented in each diet to obtain isoprotein diets. The ingredients were weighed individually (Radwag®), after being mixed in a semi-industrial mixer (Venâncio®).

The experimental groups were: SF: Ferrous sulfate; CBCC: conventional beans + conventional cassava; CBBC: conventional beans + biofortified cassava; BBCC: biofortified beans + conventional cassava; BBBC: biofortified beans + biofortified cassava. In all groups, conventional rice was added, except for the ferrous sulfate group (Table 1).

**Table 1.** Composition of the diets' tests.

| FOOD | DEPLETION | REPLETION(g/1000 g) | | | | |
|---|---|---|---|---|---|---|
| | | SF [1] | CBCC [2] | CBBC [3] | BBCC [4] | BBBC [5] |
| Ferrous sulfate | 0 | 0.1495 | 0 | 0 | 0 | 0 |
| Conventional beans | 0 | 0 | 253 | 253 | 0 | 0 |
| Biofortified Beans | 0 | 0 | 0 | 0 | 216 | 216 |
| Rice | 0 | 0 | 214 | 214 | 182 | 182 |
| Biofortified Cassava | 0 | 0 | 0 | 96 | 0 | 82 |
| Conventional cassava | 0 | 0 | 96 | 0 | 82 | 0 |
| Albumin | 200 | 200 | 80 | 80 | 87 | 87 |
| Maltodextrin | 132 | 132 | 132 | 132 | 132 | 132 |
| Sucrose | 100 | 100 | 100 | 100 | 100 | 100 |
| Soybean oil | 70 | 70 | 70 | 70 | 70 | 70 |
| Cellulose | 50 | 50 | 0 | 0 | 0 | 0 |
| Mineral mix 0% Iron | 35 | 35 | 35 | 35 | 35 | 35 |
| Vitamin mix | 10 | 10 | 10 | 10 | 10 | 10 |
| L-Cystine | 3 | 3 | 3 | 3 | 3 | 3 |
| Choline bitartrate | 2.5 | 2.5 | 2.5 | 2.5 | 2.5 | 2.5 |
| Starch | 397.5 | 397.5 | 4.5 | 4.5 | 80.5 | 80.5 |

[1] SF: Ferrous sulfate; [2] CBCC: Conventional beans+ conventional cassava; [3] CBBC: Conventional beans + biofortified cassava; [4] BBCC: Biofortified beans + conventional cassava; [5] BBBC: Biofortified beans + biofortified cassava.

*2.5. Bioavailability*

2.5.1. Animal Assay

For the biological assay, 40 male Wistar rats from the Central Animal Breeding of the Federal University of Espirito Santo (UFES), with an average initial weight of 70 g, were used. The hemoglobin depletion/repletion method was performed, following the AOAC [16], with adaptation to 21 days of depletion and 14 days of repletion. The animals were kept individually in stainless-steel cages, with a temperature of 23 °C ± 1 °C and a photoperiod of 12 h. The study was approved by the Committee on Ethics in the Use of Animals (CEUA/UFES), Protocol No. 72/2016.

2.5.2. Depletion/Repletion Method

The animals received the iron-free depletion diet for 21 days and ultra-pure water ad libitum. After this period, hemoglobin was quantified, blood (10 µL) was collected by the tail end for hemoglobin determination, and the anemic animals were divided into 5 groups (*n* = 8), according to the hemoglobin concentration, so that the mean groups were as close as possible. In the repletion phase, the animals of each group received the test diets (12mg Fe/kg of diet), and in the control group, the iron content was derived from the ferrous sulfate. At this stage, the animals were kept on their experimental diets for 14 days and received ultra-pure water ad libitum. Dietary intake was controlled, providing approximately 18 g of diet per day. At the end of this phase, hemoglobin was determined, and the hemoglobin gain was calculated by the difference of the values obtained at the end of the repletion and depletion phases.

At the end of the repletion phase, the animals were anesthetized with 0.2 mL anesthetic/100 g of animal, containing 2% xylazine hydrochloride (Syntec) and 10% ketamine hydrochloride (Syntec). The liver and 5cm of the upper small intestine immediately below the stomach of the animals were collected for determination of biomolecular analysis. The relative bioavailability values (RBV) of iron in test foods were calculated considering the standard (ferrous sulfate) with bioavailability equal to 100%.

2.5.3. Hemoglobin

For the determination of hemoglobin (Hb), the cyanmethemoglobin method [25] was used using commercial Bioclin® kits.

2.5.4. Efficiency in Hemoglobin Regeneration (HRE)

It was calculated using the formula [26]:

$$\text{HRE (\%)} = (100 \times (\text{mg Fe of final Hb} - \text{mg Feof initial Hb}))/\text{consumed Fe.} \tag{1}$$

Final Hb = Final hemoglobin; Initial Hb = Initial hemoglobin

The iron content in hemoglobin was estimated:

$$\text{Iron content} = (\text{body weight (g)} \times \text{Hb (g/L)} \times 0.335 \times 6.7)/1000. \tag{2}$$

This variable was calculated assuming that the total blood volume equals 6.7% of the body weight, and the body iron in the hemoglobin content was 0.335 g/L.

The relative biological value of hemoglobin regeneration (HRE) (RBV-HRE) was calculated as follows:

$$\text{RBV-HRE} = \text{HRE of each animal/mean of HRE of positive control} \tag{3}$$

*2.6. Biomolecular Analysis*

2.6.1. Extraction of Total Ribonucleic acid-RNA of Duodenal Mucosa and Liver

Total RNA extraction was performed using Trizol reagent (Invitrogen Brazil Ltd). Approximately 100 mg of liver and intestine were weighed, continuously macerated in liquid nitrogen, and added with 1 mL of Trizol. The homogenate was transferred to a microtube, incubating it for 5 min at room temperature, and stirring. After lysis of the cells, the RNA was extracted with 200 μL of chloroform (Sigma-Aldrich Brazil Ltd.) and centrifuged at $12{,}000 \times g$ for 15 min at 4 °C. The aqueous phase was then collected and added to 500 μL of isopropanol (Sigma-Aldrich Brazil Ltd) and incubated at room temperature for 10 min, stirring every three minutes.

The RNA was then precipitated by centrifugation at $12{,}000 \times g$ for 10 min. The supernatant was discarded, and the pellet was washed with 75% ethanol, and then centrifuged at $9500 \times g$ for 5 min. The 75% ethanol was then discarded, and the pellet was incubated at room temperature for evaporation of all ethanol. Finally, the pellet was resuspended in 50 μL of DEPC-treated water (Mili-Q water treated with 0.01% diethyl pyrocarbonate, Invitrogen Brazil Ltd, autoclaved twice) and stored at −80 °C. RNA samples were quantified by spectrophotometry using the μDrop Plate accessory (Thermo Scientific Multiskan GO model). The spectrophotometer was previously zeroed for absorbance with 2 μL of water (white), then 2 μL of the RNA samples (diluted in 1:10 with DEPC water) were applied to the nanocell, obtaining the absorbance values in two wavelengths: 260 nm and 280 nm.

The RNA concentration as well as the absorbance and the ratio between A260 and A280 were automatically expressed in the spectrophotometer. The RNA concentration is obtained as follows:

$$\text{RNA (μg/μL)} = \text{A} \times \text{A260} \times \text{dilution} \tag{4}$$

where A is a constant of each substance and is defined as the intrinsic capacity of the analyzed material to absorb light over a given length of wave. In the case of RNA, the value of this constant is 40. A260 is the reading of the absorbance of the sample at the wavelength of 260 nm. Dilution corresponds to the number of times the sample was diluted to be read.

2.6.2. Complementary DNA (cDNA) Synthesis

The cDNA synthesis was performed using the M-MLV Reverse transcriptase kit (Moloney Murine Leukemia Virus Reverse Transcriptase, Invitrogen Brazil Ltd.). Initially, 2 μg of RNA were treated with 2 μL of DNAse (Deoxyribonuclease, Invitrogen Brazil Ltd.), and 1 μL of buffer, completing the volume with DEPC water up to 10 μL. The mixture was incubated for 15 min at room temperature for the action of the DNAse. Then, 1 μL of EDTA (Ethylenediamine tetraacetic acid) was added for

inactivation of the enzyme, leaving the reaction for 8 min in a water bath at 65 °C and then on ice. The mixture was added to 1 µL of oligo dT at 100 µM and 1 µL of 10 µM dNTPs (Deoxyribonucleotide triphosphate, Invitrogen Brazil Ltd.), and then incubated for 5 min at 65 °C.

The mixture was again placed on ice and 7 µL of the mix (4 µL 5× FF buffer, 2 µL 0.1M Dithiothreitol-DTT and 1 µL Recombinant Ribonuclease Inhibitor - RNAse Out) were added. Then, the mixture was placed in a water bath at 37 °C for 1 h, and 1 µL of M-MLV reverse transcriptase was added in each reaction. To inactivate the enzyme, the cDNA was placed at 70 °C for 10 min. The cDNA was quantified by spectrophotometry using the µDrop Plate accessory (Thermus Scientific Multiskan GO Model). Then, the cDNA was stored at −80 °C until the real-time polymerase chain reaction–RT-PCR amplification reaction.

### 2.6.3. Quantification of Transcripts by Reverse Transcription Followed by Real-time polymerase chain reaction RT-PCR

The AB StepOne Real Time PCR System (Applied Biosystems) was used to perform the experiment. The relative quantification by the 2-ΔΔCt method described by Livak [27] was analyzed. Detection was performed by the SYBR$^{TM}$ Green Master Mix Reagent (Applied Biosystems), with the glyceraldehyde-3-phosphate dehydrogenase (GAPDH) enzyme gene being used as the internal control for quantification of the target gene. The samples were analyzed in two biological replicates and quantified in independent runs, with each sample being evaluated in duplicate in each reaction plate. The components for each reaction were 2 µL of cDNA, 5.0 µL of 2× SYBR Green Master Mix (Applied Biosystem), and primers at 400 nM concentrations.

The efficiency test was performed at serial dilutions of 1:10 cDNA. The reactions were prepared to obtain a final volume of 0.8 µL, consisting of: 2 µL of the cDNA; 2.4 µL of the 2.5 µM primer mixture (sense and antisense), and 5 µL of the SYBR Green reagent. Negative controls (NTC) were made by substituting cDNA samples for the same volume of water in the reaction. The amplification conditions for all genes were: at 95 °C for 20 min, 40 cycles of denaturation at 95 °C for 30 s, and annealing and extension at 60 °C for 30 s. After 40 cycles of amplification, all samples were submitted to gradual denaturation to elaborate the dissociation curve.

The samples were heated until reaching the limit of 95 °C. Primer sequences (Table 2) were used to amplify DMT-1 (divalent metal transporter), DcytB (duodenal cytochrome B), ferroportin, and hephaestin in the duodenum. In the liver, ferritin and transferrin genes were analyzed. Relative mRNA expression levels were normalized by endogenous glyceraldehyde 3-phosphate dehydrogenase (GAPDH) control. All steps were performed using free RNase conditions.

**Table 2.** Primers Used in Real-Time PCR.

| Name | Primer 1 (5′–3′) | Primer 2 (3′–5′) |
| --- | --- | --- |
| GAPDH [1] | AGGTCGGTGTGAACGGATTTG | TGTAGACCATGTAGTTGAGGTCA |
| Ferritin | CAGCCGCCTTACAAGTCTCT | ATGGAGCTAACCGCGAAGAC |
| Transferrin | AGCTGCCACCTGAGAACATC | CGCACGCCCTTTATTCATGG |
| Ferroportin | TTCCGCACTTTTCGAGATGG | TACAGTCGAAGCCCAGGACCGT |
| DMT-1 [2] | CTGATTTACAGTCTGGAGCAG | CACTTCAGCAAGGTGCAA |
| Hephaestin | GGCACAGTTACAGGGCAGAT | AGTAACGTGGCAGTGCATCA |
| DcytB [3] | TGCAGACGCAGAGTTAAGCA | CCGTGAAGTATACCGGCTCC |

[1] GAPDH: Glyceraldehyde 3-phosphate dehydrogenase; [2] DMT-1: Divalent metal transporter-1 protein; [3] DcytB: Duodenal cytochrome B.

### 2.7. Statistical Analysis

The animal groups ($n = 8$) and biomolecular analysis ($n = 5$) were compared using analysis of variance (ANOVA), followed by the Tukey test ($p < 0.05$). Statistical analysis was done with SPSS Statistics Data Editor version 19.0 (IBM SPSS Statistics Base, DMSS, SP, Brazil).

## 3. Results

The chemical composition of the flours of the raw materials is shown in Table 3.

**Table 3.** Chemical composition of test food (100 g).

| *n* = 3 | CB [1] | BB [2] | CC [3] | BC [4] | Rice [5] |
|---|---|---|---|---|---|
| **Energy (kcal)** | 294.40 ± 0.70 | 269.52 ± 0.89 | 315.50 ± 1.12 | 231.72 ± 0.26 | 371.11 ± 1.48 |
| **Carbohydrate (g)** | 44.62 ± 0.82 | 34.48 ± 0.43 | 75.48 ± 0.37 | 53.23 ± 0.29 | 81.23 ± 0.70 |
| **Protein (g)** | 24.80 ± 0.86 | 28.64 ± 0.53 | 3.13 ± 0.01 | 4.03 ± 0.03 | 11.48 ± 0.40 |
| **Lipid (g)** | 2.00 ± 0.05 | 1.90 ± 0.17 | 0.12 ± 0.28 | 0.30 ± 0.11 | 0.03 ± 0.05 |
| **Ashes (g)** | 3.27 ± 0.08 | 3.51 ± 0.13 | 2.23 ± 0.05 | 2.45 ± 0.07 | 0.37 ± 0.08 |
| **Moisture (g)** | 7.54 ± 0.07 | 6.14 ± 0.05 | 10.19 ± 0.09 | 10.00 ± 0.20 | 6.54 ± 0.45 |

[1] CB: Conventional cowpea beans; [2] BB: Biofortified cowpea bean; [3] CC: Conventional cassava; [4] BC: Biofortified cassava; [5] Conventional rice.

Table 4 shows iron content, *β*-carotene, phenolic compounds, phytic acid, phytate/iron molar ratio, and fiber raw materials.

**Table 4.** Content of phenolic compounds, phytate, and fibers.

| *n* = 3 | CB [1] | BB [2] | CC [3] | BC [4] | Rice [5] |
|---|---|---|---|---|---|
| **PHYTATE (g/100g)** | 0.21 ± 0.18 | 0.29 ± 0.13 | 0.18 ± 0.05 | 0.06 ± 0.02 | 0.02 ± 0.00 |
| **IRON (mg/kg)** | 43.83 | 52.41 | 5.96 | 12.33 | 1.94 |
| **PHYTATE/IRON MOLAR RATIO** | 4.11 | 4.73 | 25.41 | 4.15 | 6.98 |
| **β-CAROTENE (μg/g)** | - | - | 0.9 | 7.6 | - |
| **PHENOLIC COMPOUNDS (mg gallic acid/100g)** | 0.04 ± 0.00 | 1.42 ± 0.05 | 0 ± 0.00 | 0.41 ± 0.1 | 0 ± 0.00 |
| **TOTAL FIBER (g/100g)** | 18.09 | 25.34 | - | - | 0.36 |
| **INSOLUBLE FIBER (g/100g)** | 15.13 | 19.91 | - | - | 0.00 |
| **SOLUBLE FIBER (g/100g)** | 2.96 | 5.43 | - | - | 0.36 |

[1] CB: Conventional cowpea beans; [2] BB: Biofortified cowpea bean; [3] CC: Conventional cassava; [4] BC: Biofortified cassava; [5] Conventional rice. Phytate/iron molar ratio = ((mg phytate/100g)/660)/((mg iron/100g)/55.85).

The iron contents of 43.83 and 52.41 g/kg were found in conventional and biofortified cowpea beans, respectively. This increase on iron content was below the target for biofortification, although it was 19.6% higher than the conventional, which may contribute to reach the daily recommended intake of iron.

Although the phytate content did not significantly differ among the cowpeas, the phytate/iron molar ratio of the biofortified cowpea was 15% higher than the conventional. Also, the phenolic content did not differ between them, but the content on the biofortified cowpea was 1.42 mg/100 g, compared to 0.04 mg/100 g on the conventional.

The content of *β*-carotene found was considerably high in biofortified cassava at the concentration of 7.6 μg/g.

### 3.1. Animal Assay

The group that consumed biofortified beans and cassava showed a significant increase in body weight gain, compared to the other groups (Table 5).

There was no significant difference between the control and test groups for food consumption. The diets were formulated to provide 12mg/kg of iron, according to AOAC [16]. It is of great importance to perform the hemoglobin regeneration efficiency, which will consider iron consumption and hemoglobin gain (Table 6) of the animals in each group.

**Table 5.** Weight gain, food consumption, and food efficiency ratio (FER) of experimental groups.

| EXPERIMENTAL GROUPS (*n* = 8) | Weight Gain (g) | Food Consumption (g/day) | FER [6] |
|---|---|---|---|
| SF [1] | 45.41 ± 12.89 [b] | 16.17 ± 1.43 [a] | 0.20 ± 0.04 [b] |
| CBCC [2] | 59.66 ± 13.74 [b] | 16.18 ± 2.15 [a] | 0.26 ± 0.04 [a] |
| CBBC [3] | 57.90 ± 15.95 [b] | 16.91 ± 2.46 [a] | 0.24 ± 0.05 [ab] |
| BBCC [4] | 57.66 ± 8.13 [b] | 16.61 ± 1.81 [a] | 0.25 ± 0.02 [ab] |
| BBBC [5] | 79.54 ± 14.92 [a] | 17.63 ± 2.05 [a] | 0.28 ± 0.05 [a] |

[1] SF: Ferrous sulfate; [2] CBCC: Conventiona beans+conventional cassava; [3] CBBC: Conventional beans+biofortified cassava; [4] BBCC: Biofortified beans+conventional cassava; [5] BBBC: Biofortified beans+biofortified cassava; [6] FER: Food efficiency ratio. Means followed by different letters in the column are statistically different by Tukey test (*p* < 0.05).

**Table 6.** Initialhemoglobin (Hb), final Hb, and Hb gain in the repletion phase.

| EXPERIMENTAL GROUPS (*n* = 8) | Initial Hb [6] (g/dL) | Final Hb [6] (g/dL) | Hb [6] Gain (g/dL) |
|---|---|---|---|
| SF [1] | 6.25 ± 0.56 | 8.04 ± 0.84 | 1.79 ± 1.04 |
| CBCC [2] | 6.25 ± 0.52 | 7.98 ± 1.04 | 1.73 ± 1.13 |
| CBBC [3] | 6.24 ± 0.53 | 7.99 ± 0.70 | 1.75 ± 0.74 |
| BBCC [4] | 6.26 ± 0.47 | 8.10 ± 1.18 | 1.83 ± 1.14 |
| BBBC [5] | 6.29 ± 0.46 | 8.00 ± 0.93 | 1.72 ± 0.78 |

[1] SF: Ferrous sulfate; [2] CBCC: Conventional beans+conventional cassava; [3] CBBC: Conventional beans+biofortified cassava; [4] BBCC: Biofortified beans+conventional cassava; [5] BBBC: Biofortified beans+biofortified cassava; [6] Hb: Hemoglobin. Means did not differ between them in the column (*p* > 0.05).

The hemoglobin concentration of the depletion phase did not differ significantly between the groups, since the animals were grouped to maintain the hemoglobin mean homogeneous. All animals recovered partially from the anemia caused during the depletion period. Thus, the relationship between the hemoglobin gain and the iron consumed was verified (Table 7).

**Table 7.** Iron consumed, hemoglobin gain/iron consumed, hemoglobin regeneration efficiency (HRE), and relative bioavailability value (RBV of HRE).

| DIETS (*n* = 8) | Fe [6] Consumed (mg) | Hb [7] Gain (g/dL)/Fe Consumed (g) | HRE [8] (%) | RBV [9] of HRE (%) |
|---|---|---|---|---|
| SF [1] | 2.72 ± 0.24 | 0.65 ± 0.36 | 51.02 ± 19.34 | 100.00 |
| CBCC [2] | 2.72 ± 0.36 | 0.67 ± 0.48 | 57.96 ± 14.25 | 113.60 |
| CBBC [3] | 2.84 ± 0.41 | 0.61 ± 0.25 | 57.17 ± 13.10 | 112.05 |
| BBCC [4] | 2.79 ± 0.30 | 0.69 ± 0.47 | 60.54 ± 17.27 | 118.67 |
| BBBC [5] | 2.96 ± 0.34 | 0.57 ± 0.25 | 63.97 ± 18.17 | 125.39 |

[1] SF: Ferrous sulfate; [2] CBCC: Conventional beans+conventional cassava; [3] CBBC: Conventional beans+biofortified cassava; [4] BBCC: Biofortified beans+conventional cassava; [5] BBBC: Biofortified beans+biofortified cassava; [6] Fe: Iron; [7] Hb: Hemoglobin; [8] HRE: Hemoglobin regeneration efficiency; [9] RBV: Relative bioavailability value. Means did not differ between them in the column (*p* > 0.05)

*3.2. Biomolecular Analysis*

The BBCC group (6.28 ± 2.68) had greater liver gene expression of ferritin (Figure 1a) and the BBBC group (0.67 ± 0.5) had lower gene expression.

The transferrin transporter (Figure 1b) was less expressed in the groups containing conventional cowpea, CBCC (0.27 ± 0.19) and CBBC (0.33 ± 0.18), suggesting lower iron release to be absorbed in enterocytes. This result was compatible with the bioavailability found in this study, which had lower absorption, although not significant in the groups with conventional cowpea, regardless of the vitamin A presence.

There was no significant difference in mRNA expression of DcytB (Figure 2a).The iron transporter in the enterocyte, DMT-1, also did not differ between groups (Figure 2b), although the groups with

biofortified cassava, CBBC (1.69 ± 2.67) and BBBC (2.06 ± 0.99), presented numerically superior values to the control group and the groups that contained conventional cassava.

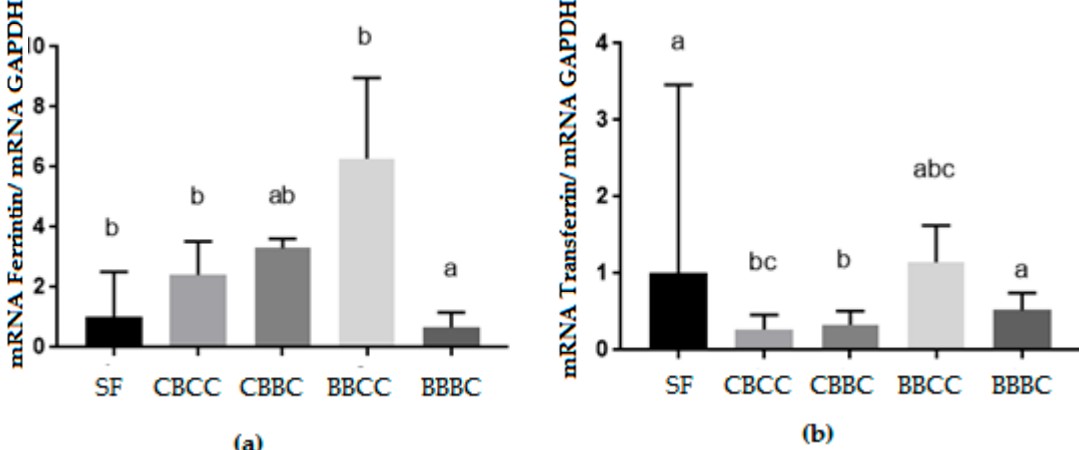

**Figure 1.** (**a**) Biomolecular analysis of ferritin mRNA, (**b**) Biomolecular analysis of transferrin mRNA. SF:Ferrous sulfate; CBCC:Conventional beans+conventional cassava; CBBC: Conventional beans+biofortified cassava; BBCC: Biofortified beans+conventional cassava; BBBC: Biofortified beans+biofortified cassava. Means followed by the same letter in the bars do not differ from each other ($p > 0.05$).

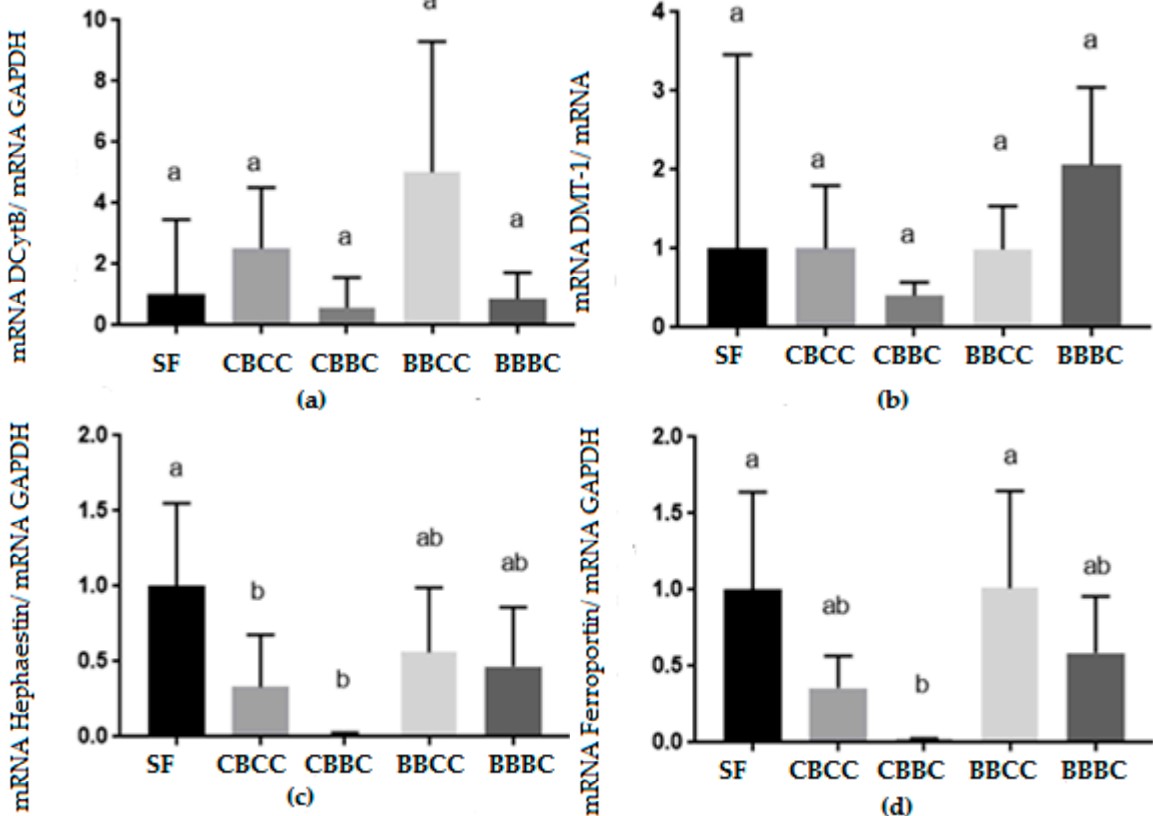

**Figure 2.** (**a**) DcytB mRNA; (**b**) DMT-1 mRNA; (**c**) Hephaestin mRNA; (**d**) Ferroportin mRNA. SF: Ferrous sulfate; CBCC:Conventional beans+conventional cassava; CBBC: Conventional beans+biofortified cassava; BBCC: Biofortified beans+conventional cassava; BBBC: Biofortified beans + biofortified cassava. Means followed by the same letter in the bars do not differ from each other ($p > 0.05$).

There was no significant difference between the groups fed with cowpea beans, regardless of the presence of biofortified cassava, as for the mRNA of hephaestin (Figure 2c). However, the groups containing biofortified beans, BBCC (0.56 ± 043) and BBBC (0.47 ± 0.39), did not differ from the control (Ferrous sulfate).

As in hephaestin, in the biomolecular analysis of ferroportin (Figure 2d), the groups with biofortified beans were more expressed, BBCC (1.01 ± 0.64) and BBBC (0.59 ± 0.37), carrying higher iron content of the enterocytes into the bloodstream, confirming the greater bioavailability of these groups.

The biofortified cowpea (BRS *Arace*) presented an increase of approximately 19.5% in iron content in relation to conventional cowpea (BRS *Nova Era*). In parallel, the biofortified cowpea had higher levels of phytates, phenolic compounds, and fibers in relation to conventional cowpea.

Mucosa of rats fed biofortified cowpea with iron was similar to the mucosa of rats fed ferrous sulfate regarding the expression of the hephaestin and ferroportin proteins, suggesting a greater efficiency in the intestinal absorption of iron. On the other hand, the expression of transferrin in the groups fed conventional cowpea was lower when compared to the control group, ferrous sulfate, which indicates lower absorption of iron in these groups. The expression of ferritin in the liver was lower in the group fed with both biofortified foods, BBBC, which may indicate a greater mobilization of hepatic iron.

## 4. Discussion

There was great variation regarding protein, carbohydrate, and moisture contents between the two types of cowpeas (Table 3). The content of protein, lipid, ashes, and moisture found in the beans were consonant to those found in the literature. Brigide [28] found, in irradiated beans, 23.9 g/100 g of protein. A study by Bigonha [29] evaluated biofortified BRS Agreste and Pontal varieties and found protein levels of 23.40 g/100 g and 21.01 g/100 g, respectively. Vaz-Tostes et al. [30], in BRS Pontal (biofortified) and Perola (conventional) varieties of beans, found protein levels of 18.36 g/100 g and 21.51 g/100 g, respectively.

Variations in the compositions may occur because of the location in which they were grown and the environmental conditions. According to a study by Marinho, Pereira, and Costa [31], cowpea beans have a slightly higher protein content than common beans (*Phaseolus vulgaris*), varying from 20% to 30%. It is important to evaluate the protein content, as it may increase the bioavailability of iron due to the need of iron transporters during absorption [32].

There was no great variation in protein, lipid, ash, and moisture contents between the two types of cassava (Table 3). The protein content found in cassava flour was higher than the studies of Feniman [33] (0.70 g/100 g) and Dias and Leonel [34] (0.71 g/100 g). There was variation between the carbohydrate contents: the conventional cassava presented a higher content in relation to the biofortified one, such difference is due to the high total fiber content of the biofortified cassava (Table 4).

Tako et al. [35] found higher contents of iron, 78.8 mg/kg iron in red beans, whereas Correa [36] found results similar to this study, with 57.8 mg/kg in biofortified *Aracê* cowpea. In addition to the antinutritional factors, there may be a change in the composition of the food, due to the cooking process, reducing the content of minerals such as iron and zinc [37].

Vaz-Tostes et al. [30] used conventional and biofortified beans to evaluate the bioavailability of iron and found iron contents similar to this study, 60.52 and 52.43 mg/kg of Fe, biofortified and conventional, respectively. Different from Pachón et al. [38], that between biofortified and conventional beans does not present significant difference.

Mezette et al. [39] found a similar result in cooked cassava roots, with iron contents of 6.2 to 10.9 mg/kg, the variation occurred due to the different genotypes. A component that reduces the bioavailability of iron is phytate, which acts to form insoluble complexes and, thereby, decreases absorption [40].

Tako et al. [41] also found higher phytate content in biofortified beans, but due to the higher amount of iron in the biofortified ones, the phytate:iron ratio was higher in conventional beans [42].

The phytate:Fe molar ratio indirectly evaluates the bioavailability of iron: when the result is greater than 1, it means that there may be low bioavailability. In this study, all foods were greater than 1. For the studied raw material, the values were between 4 and 7, except for conventional cassava, where the result was 25.41. The diets containing biofortified cassava (CBBC) and (BBBC), had a lower phytate/Fe molar ratio, 4.22 and 4.68respectively, in relation to diets containing conventional cassava (CBCC, 6.67) and (BBCC, 6.84). Thus, lower interference in the bioavailability of iron in diets with higher carotenoid content. According to Correa [36], the molar ratio found for conventional and biofortified cowpea beans suggests that phytate may compromise the bioavailability of iron.

Another factor that negatively influences iron absorption is the phenolic compounds. One of the factors that increase the content of these compounds is the maceration and the cooking of the grains, because with the solubility of the phenolic compounds, they are released [43].In addition to the iron content, its bioavailability should be considered, and may be influenced by several factors, mainly by other food components present in the diet [32,44].Fibers can also exert negative effects on the absorption of iron, it can bind to the ions of minerals and prevent this absorption. In addition, the fiber, by exerting influence of intestinal transit and decreasing the time, can reduce the absorption of minerals [45].

The content of β-carotene found in this study was similar to the results obtained by EMBRAPA with a content of 8.73 μg/g [46], possibly due to a greater presence of trans β-carotene, which is predominant in biofortified cassava [47]. Berni et al. [47] found similar values, 6.4 μg/g, even in cooked biofortified cassava, with the same processing performed in the raw material of this study.Silva et al. [48] conducted a study with conventional *Cerrado* cassava, finding a value of 2.31 and 3.40 μg/g respectively, in the *Pretinha* and BRS *Dourada* varieties, respectively. In comparison, the β-carotene content used as a reference in this study was 9 μg/g, a target of the cultivars of the BioFORT biofortification network. Reduction of pro-vitamin A carotenoids may be a consequence of the exposure time on the heat treatment and the presence of oxygen to which they were subjected, such as drying [49].

The group that consumed biofortified beans and cassava showed a significant increase in body weight gain, compared to the other groups (Table 5). This difference shows that the biofortified food diet meets the needs of animals, providing a more effective weight gain. According to Toaiari et al. [50], this difference in weight gain may be due to the bioavailability of iron in the diets' tests, which is requested for young animal growth. Iron deficiency, resulting from the long period of depletion, can lead to an absorption difficulty, resulting in less weight gain in animals [51]. Food consumption of the BBBC group was similar to the other groups, but FER was higher than the control ferrous sulfate group, which means a better conversion of the energy consumed intobody weight gain in animals fed biofortified foods [52].

The literature shows that vitamin A helps iron absorption, but in this study, there was no significant difference ($p < 0.05$) in hemoglobin gain per iron consumed in the different groups. The efficiency of hemoglobin regeneration showed no significant difference between the groups, regardless of the presence of cassava (Table 7). Bigonha [29] evaluated that bioavailability of the cultivars BRS Agreste (84.6 mg/kg) and BRS Pontal (96.4 mg/kg) biofortified beans was not significant in relation to the control group (ferrous sulfate), demonstrating efficiency of the beans in the hemoglobin recovery.

One factor that positively influences the evaluation of iron bioavailability in rats is the ability to synthesize ascorbic acid, which may lead to higher absorption of iron. Another factor is the presence of the enzyme phytase, which increases the bioavailability of minerals, due to the capacity to undo the phytate chelates with minerals. Although this animal model may overestimate the bioavailability, it is favorable because of the easy handling and allocation, besides the low cost, as well as the aid in the definition of genotypes that can be improved in biofortification programs [53].

Murray-Kolb et al. [54] used the depletion/repletion method in piglets, with diets based on white and red beans, and they did not obtain a significant difference in the final concentration of hemoglobin between the groups. Arruda et al. [13] observed that diets deficient in vitamin A for 57 days influenced lower weight gain and higher concentration of hemoglobin in animals.

The BBCC group (6.28 ± 2.68) demonstrates that there was a greater accumulation of iron in the organism, possibly associated with the lower pro-vitamin A carotenoid content of the conventional cassava (Figure 1a). On the contrary, the BBBC group (0.67 ± 0.5) had lower gene expression and may be associated with vitamin A activity in the release of iron from the hepatic reserve (Figure 1a). According to Martini [55], there is an inverse relationship in the presence of retinol and iron in the liver Ferritin is responsible for iron storage, when there is iron excess in the cells, being able to avoid oxidative damages.

Dias et al. [56], when evaluating the ferritin gene expression, found lower expression of ferritin in the liver of animals fed with iron-biofortified beans (BRS Pontal), without addition of pumpkin or sweet potato biofortified with beta-carotene, contrasting the results of this study. They also found greater expression of DcytB in the group that contained biofortified beans associated with vitamin A, contrasting the findings of this study.

The iron transporter in the enterocyte, DMT-1, also did not differ between groups (Figure 2b). It is possible that the higher carotenoid content may have aided in the mobilization of iron, as pointed out in the studies that affirm the role of vitamin A in iron metabolism [57–59]. Moreira [60] did not find significant differences between the groups that were treated with 35mg of a Fe/kg diet with and without vitamin A, suggesting that vitamin A does not act directly at the levels of DMT-1.

The hephaestin of the groups containing biofortified cowpeas, BBCC (0.56 ± 043) and BBBC (0.47 ± 0.39) (Figure 2c), did not differ significantly from the control, so it is assumed that ferrous iron was converted to ferric in the basolateral membrane, resulting in greater iron uptake in these groups. In the study by Dias et al. [56] on the other hand, the presence of carotenoids in the bean diet promoted greater gene expression of hephaestin, observing the influence of vitamin A on iron absorption.

Ferroportin was more expressed in the groups fed with biofortified cowpeas, BBCC (1.01 ± 0.64) and BBBC (0.59 ± 0.37) (Figure 2d). Silva [10] found lower expression of ferroportin in the group that contained Chia, responsible for its higher iron content. In the study by Dias et al. [56], the higher pro-vitamin A carotenoid content in the diet influenced the induction of ferroportin.

The expression of these proteins depends on the stage of depletion of the animal, the iron intake, and its bioavailability. Therefore, the values may have been lower because the analyses are done at the end of the experiment, when the animals already have hemoglobin levels recovered, even if these proteins are partially associated with the iron deficiency compensation in the body. As the animals were not anemic in the collection of organs, they may not report the effective function of these transporters [56].

## 5. Conclusions

This study points out a higher efficiency of fortified cowpea in the bioavailability of iron, regardless of the presence of vitamin A. However, new studies are necessary to verify the influence of vitamin A on iron absorption, considering vitamin A provided contents in the diet and factors that may positively or negatively influence this interaction and absorption.

**Author Contributions:** P.T.A., C.T.S., and R.A.d.F. carried out the experiment. P.T.A. took the lead in writing the manuscript. M.d.G.V.-T. contributed to interpretation of results and statistical analysis. R.C.L.T. carried out the biomolecular analysis and writing. N.M.B.C. conceived the idea and was in charge of overall direction of the project and manuscript corrections.

**Funding:** FAPES – Fundação de Amparo à Pesquisa e Inovação do Espirito Santo, Grant # 03/2017 Universal, TO 201/2017.

**Conflicts of Interest:** The authors declare no conflict of interest.

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
