# Peer review of "Bioavailability of Iron and the Influence of Vitamin a in Biofortified Foods"

_agronomy, doi:10.3390/agronomy9120777_

Round 1

Reviewer 1 Report

Dear Authors,

The work is interesting, but many parts and data are confusing. Specific comments are presented below:

Abstract

Line 13: not only habits lead to nutritional deficiencies.

Line 16: correct : pro-vitamin instead of po-vitamin

Introduction

Line 37 – mistake: thTave? 

Line 44 – genetic breeding – thought abbreviation, sometimes used, but better – breeding of genetically modified plants.

Materials and Methods

Lines 58-61

You should more emphasize that cassava was biofortified in pro-VitA – not rice. Now it is not clear.

Please, rewrite the composition of the diet. It is very difficult to understand this part. Please divide it into: 1. depletion diet description; 2. test (experimental) diets. Of course, table is OK, but description is not clear.

Moreover, I am not sure which was the content of iron in the depletion diet? It is not  possible to obtain diet free of iron (because Fe contaminate EVERYTHING).

At first I’ve understood, that depletion diet contained 12 ppm of Fe – because AIN-93G diet recommend 52.02 ppm for mice and rats. Thus – which diet contained 12 ppm? It is too small amount for experimental and too high for depletion (if you want to anaemise animals in 21 days). It is the weakest point of the manuscript. Undermines the reliability of presented results.

Lines 162 and 164 - TRIzol or trizol?

Describe which is the aim to analyze the expression of these gens – in the introduction e.g.

Results and discussion

Line 237 – or in another place of the manuscript; the increase in iron content in fortified raw material is not high. I think, that you should introduce information about the way in which biofortified seeds were obtained - what kind of modification it was. It maybe also introduced into materials and methods part.

Line 239 – please add and information that phytate/iron molar ratio differed ~15%

Line 248-268 - the same problem – as it was noted in material and method. In my opinion it is not possible to observe such high increase in Hgb content after feeding animals  for 2 weeks with the diet containing 12 mg of iron (from ~6.3 up to ~8 g/dL). The content of iron is more than four times less than recommended value (52.02 ppm). Thus,  if the rats still got an iron deficiency diet (the diet containing 12 ppm is a deficiency diet), I am not sure if it was possible to observe such a high weight gain, as it was noted in Tab. 5.

Tab. 7. Moreover, if a rat consumed ~2.8 mg of iron per day, it means that they consumed ~230 g of fodder (if tested diet contains 12 mg of iron). Authors have started with rats ~70 g, than they were anaemised – 21 days (weight gain should be strongly limited). Next, they consumed 230 g of fodder per day?? How is it possible? It is more than their weight.

I am also not sure if presented HRE is important parameter. You should be focused on IDA preventing – not in weight gain. The rats with higher HRE still are deficient in iron – the aim of their nutrition is not achieved. Concentration of Hgb 8 g/dL it is still anaemia state. Of course, result may be presented, but it should be discussed less enthusiastic in the rest of the manuscript.

Still Tab. 7.

correct caption;

check statistical significance of the observed differences

lines 269-294

Why BBCC group had higher expression of ferritin? If you cannot explain it – please inform. But the differences on BBCC and BBBC for me are surprising.

Line 298

Such shortcuts should not be used: ‘biofortified cowpea…was similar…regardin the expression…..”

It should be: Mucosa of rats fed with biofortified with iron bean was similar to the mucosa of rats fed with diet containing FeSO4….

The same - in line 300

Line303: greater mobilization of hepatic iron may be very dangerous. It informs  about decreasing of storage iron content. That is why I have written that increase in HRE is not important if it does not take place together with Hgb concentration  increase.

Discussion

310-320- Difference in the protein content may be important in the experiment – may influence on the level of IDA.

Line 329 – is it not a mistake? conventional bean is richer in iron???!!

Line 346 – Better would be: For the studied raw materials the values were between 4-7…..

Line 346-350 – the differences in the bioavailabilty were statistically significant?

Line 356 – ‘nutrients’ not ‘food’

Line 369 – not only temperature but also oxigen!!

Line 371 -372 – was it the Authors aim? to get weight gain? I DO NOT think so. The animals with the highest weight gain were more deficient in iron (compare their Hgb concentration and ferritin expression with these parameters in others group). Of course, it is important to keep the proper weight, but it was not the aim of the study. Authors should  discussed it in another way

Line 376- the sentence here does not suit the discussion

Line 382 – the efficiency in Hgb recovery may result also from the manner of biofortification – substances formed in biofortified beans –that’s why the information about the type of fortification (and consequences) is important

Line 393 – how it was discussed? The higher Hgb and decreased weight?

Line 397 – how vit A may increase release of iron from the liver? Especially in such young organism – with small reserve of iron in the liver?

Line 401 -405 Could you explain, why Authors result could be different?

Line 407 – Mobilisation of iron could take place in the liver, I am not sure that in the lumen of intestine in the apical membrane of enterocytes. Vit A influence iron metabolism (and release), not iron uptake from the lumen intestine into the enterocytes.

Line 413- no differences on what????

Line 417 – more expressed in what????

Conclusion: the problem in the presented observation was that rats still had an IDA after the experiments. Of course – increase in Hgb (and iron content) was observed, but the state could influence on gene expression.

References: I was surprised that you have quoted almost only literature from South America. Of course – you can do it.  It is just a comment.

 After reading of the whole article I am not sure that the title is proper. Are you sure that you have checked interactions? Not: influence of pro-vit A on absorption of iron?

Author Response

Reviewer 1

Comments and Suggestions for Authors

Dear Authors,

The work is interesting, but many parts and data are confusing. Specific comments are presented below:

Abstract

Line 13: not only habits lead to nutritional deficiencies.

Response: We changed the sentence to improve the meaning:

“Inadequate eating habits, among other factors, lead to nutritional deficiencies worldwide”

Line 16: correct : pro-vitamin instead of po-vitamin

Response: 0k, correction done

Introduction

Line 37 – mistake: thTave? 

Response: mistake was corrected

Line 44 – genetic breeding – thought abbreviation, sometimes used, but better – breeding of genetically modified plants.

Response: ok, modification was done.

Materials and Methods

Lines 58-61

You should more emphasize that cassava was biofortified in pro-VitA – not rice. Now it is not clear.

Response: We changed the sentence to make it clear:

The raw material: cowpea (Vigna unguiculata L Walp), cultivar BRS Aracê, biofortified with iron and zinc,  cassava (Manihot esculenta Crantz), cultivar BRS Jari, biofortifiedwith β-carotene(pro-vitamin A) conventional cowpea (Vigna unguiculata L Walp), cultivar BRS Nova Eraandwhite commonrice (BRS Esmeralda) were provided by the Brazilian Agricultural Research Corporation (EMBRAPA). Conventional cassava was obtained from the local market in Alegre, ES, Brazil.

Please, rewrite the composition of the diet. It is very difficult to understand this part. Please divide it into: 1. depletion diet description; 2. test (experimental) diets. Of course, table is OK, but description is not clear.

Response: The sentence was rewritten:

The depletion diet was based on the AIN-93G diet [24], by using an iron-free mineral mix. The repletion diet was similar to the depletion diet, but was adjusted to provide 12mg Fe/kg diet from ferrous sulfate (FeSO4) in the control (SF) diet and from the cowpea in the test diets.

Moreover, I am not sure which was the content of iron in the depletion diet? It is not  possible to obtain diet free of iron (because Fe contaminate EVERYTHING).

Response: The depletion diet was made by adding iron-free mineral mix. The contaminating iron was not evaluated, but we used the same ingredients in the test diets also, so the contamination, if present, would be in all the diets. Besides, the depletion diet was provided to all animals prior the repletion phase irrespective to their diets in the further phase of repletion. The main reason of the depletion diet is to promote IDA in all animals.

At first I’ve understood, that depletion diet contained 12 ppm of Fe – because AIN-93G diet recommend 52.02 ppm for mice and rats. Thus – which diet contained 12 ppm? It is too small amount for experimental and too high for depletion (if you want to anaemise animals in 21 days). It is the weakest point of the manuscript. Undermines the reliability of presented results.

Response: 1.Depletion DietIn this diet, the rats received iron-free mineral mix. The content of iron (12ppm) in the test diets was provided by ferrous sulfate or cowpeas. The AOAC method of depletion-repletion indicates the use of 3 doses of iron (6, 12 and 24 ppm) in the repletion diet. This implies the use of 24 animals (n=8) in each experimental group, which increases considerably the number of animals in the experiment. The level of 6 ppm is too low to promote IDA recovery and 24 ppm is suficient to recover the deficiency, but is too high to distinguish the eficacy of each test diet, i.e, all the animals recover from IDA at the end of the trial, so the final Hb does not allow to diferenciate the groups in terms of iron bioavailability. This is the reason why we used 12 ppm. We did not intente to recover completely the IDA, otherwise we could not difetenciate the potential bioavailability of each diet. 

2.Experimental Diet

The experimental diets were adjusted to receive 12mg / kg diet. In the repletion diet the iron content was derived from the ferrous sulfate and in the test diets, different concentrations of cowpea, based on its iron content, were used to reach the stipulated iron content of 12 mg / kg diet.

Lines 162 and 164 - TRIzol or trizol?

Response: Correction done: Trizol

Describe which is the aim to analyze the expression of these gens – in the introduction e.g.

Response: We added a sentence in the Introduction to explain the role of the proteins evaluated in the iron metabolismo: The mechanisms of iron homeostasis are determined by gene expression of proteins related to iron metabolism. For instance, the duodenal cytochrome b (Dcytb) acts by reducing ferric iron (Fe3+) to the ferrous form (Fe2+), which is transported through the enterocyte membrane by the divalent metal transporter (DMT-1). The Ferroportin exports iron to plasma cells, converting ferrous iron (Fe2+) back to ferric iron (Fe3+) through hephaestin to be binded to transferrin [15].

Results and discussion

Line 237 – or in another place of the manuscript; the increase in iron content in fortified raw material is not high. I think, that you should introduce information about the way in which biofortified seeds were obtained - what kind of modification it was. It maybe also introduced into materials and methods part.

Response: We added the sentence: This increase on iron content was below the target for biofortification, although it was 19.6% higher than the conventional, which may contribute to reach the daily recommended intake of iron. .

The biofortification process is out of the scope of this manuscript, since it requires a very complex agronomic process and would be away of the main objective of this study. The focus of this manuscript is to provide nutritional information of the products developed and provides by EMBRAPA. More information can be obtained in www.embrapa.br/biofort.

Line 239 – please add and information that phytate/iron molar ratio differed ~15%

Response: we added the sentence

Although the phytate content did not significantly differ among the cowpeas, the phytate/iron molar ratio of the biofortified cowpea was 15% higher than the conventional. Also, the phenolic content did not differ between them, but the content on the biofortified cowpea was  1.42 mg/100g, compared to 0.04 mg/100g on the conventional.

Line 248-268 - the same problem – as it was noted in material and method. In my opinion it is not possible to observe such high increase in Hgb content after feeding animals  for 2 weeks with the diet containing 12 mg of iron (from ~6.3 up to ~8 g/dL). The content of iron is more than four times less than recommended value (52.02 ppm). Thus, if the rats still got an iron deficiency diet (the diet containing 12 ppm is a deficiency diet), I am not sure if it was possible to observe such a high weight gain, as it was noted in Tab. 5.

Response: As we told before, the AOAC method of depletion-repletion indicates the use of 3 doses of iron (6, 12 and 24 ppm) in the repletion diet. This implies the use of 24 animals (n=8) in each experimental group, which increases considerably the number of animals in the experiment. The level of 6 ppm is too low to promote IDA recovery and 24 ppm is suficient to recover the deficiency, but is too high to distinguish the eficacy of each test diet, i.e, all the animals recover from IDA at the end of the trial, so the final Hb does not allow to diferenciate the groups in terms of iron bioavailability. This is the reason why we used 12 ppm. We did not intente to recover completely the IDA, otherwise we could not difetenciate the potential bioavailability of each diet. 

Tab. 7. Moreover, if a rat consumed ~2.8 mg of iron per day, it means that they consumed ~230 g of fodder (if tested diet contains 12 mg of iron). Authors have started with rats ~70 g, than they were anaemised – 21 days (weight gain should be strongly limited). Next, they consumed 230 g of fodder per day?? How is it possible? It is more than their weight.

Response: We corrected the title of Table 7. The correct is the consumption of 2.8mg of iron in the total repletion period (14 days). They consumed an average of 232g of diet during this period. In Table 5 is shown the daily diet consumption of approximately 16.6g / day

I am also not sure if presented HRE is important parameter. You should be focused on IDA preventing – not in weight gain. The rats with higher HRE still are deficient in iron – the aim of their nutrition is not achieved. Concentration of Hgb 8 g/dL it is still anaemia state. Of course, result may be presented, but it should be discussed less enthusiastic in the rest of the manuscript.

Response: The objective is not completely recover the anemic animals, since their Hb levels would reach a plateau and would not differentiate the treatments in terms of iron bioavailability.

Still Tab. 7.

correct caption;

check statistical significance of the observed differences

Response: Caption and statistical significance were checked and

 corrected:  Means did not differ between them in the column (p > 0.05)

lines 269-294

Why BBCC group had higher expression of ferritin? If you cannot explain it – please inform. But the differences on BBCC and BBBC for me are surprising.

Response: This is discussed on lines 410-416: 

The BBCC group (6.28 ± 2.68) demonstrates that there was a greater accumulation of iron in the organism, possibly associated with the lower pro-vitamin A carotenoid content of the conventional cassava (Figure 1a). On the contrary, the BBBC group (0.67 ± 0.5) had lower gene expression and may be associated with vitamin A activity in the release of iron from the hepatic reserve (Figure 1a). According to Martini [56], there is an inverse relationship in the presence of retinol and iron in the liver [55]. Ferritin is responsible for iron storage, when there is iron excess in the cells, being able to avoid oxidative damages.

Line 298

Such shortcuts should not be used: ‘biofortified cowpea…was similar…regarding the expression…..”

It should be: Mucosa of rats fed with biofortified with iron bean was similar to the mucosa of rats fed with diet containing FeSO4….

The same - in line 300

Response: Corrections in both lines were done:

Mucosa of rats fed biofortified cowpea with iron was similar to the mucosa of rats fed ferrous sulfate regarding the expression of the hephaestin and ferroportin proteins, suggesting a greater efficiency in the intestinal absorption of iron. On the other hand, the expression of transferrin in the groups fed conventional cowpea was lower when compared to the control group, ferrous sulfate, which indicates lower absorption of iron in these groups.

Line303: greater mobilization of hepatic iron may be very dangerous. It informs  about decreasing of storage iron content. That is why I have written that increase in HRE is not important if it does not take place together with Hgb concentration  increase.

Response: We agree, but the mobilization is necessary for animal growth and other iron functions in the body. It may be dangerous if the anemia persists. Since the objective of this study was not to treat the IDA, it was not discussed in the manuscript.

Discussion

310-320- Difference in the protein content may be important in the experiment – may influence on the level of IDA.

Response: Yes, this may be an important issue to be evaluated in further studies concerning IDA.

Line 329 – is it not a mistake? conventional bean is richer in iron???!!

Response: We changed the sentence to:

Tako et al. [35] found higher contents of iron, 78.8 mg/kg iron in red beans, whereas Correa [36] found results similar to this study, with 57.8 mg/kg in biofortified Aracê cowpea.

Indeed, in the study of Correa, the conventional Guariba cowpea showed higher content of iron than some biofortified cowpeas. This unexpected result may be due to the soil and region where the grains were produced. This is the reason why we worked with different variety of conventional cowpea (Nova Era) and the same biofortified cowpea (Aracê) in the current study. We decided to exclude this information in the text to avoid misunderstandings.

Line 346 – Better would be: For the studied raw materials the values were between 4-7…..

Response: We changed the sentence to: For the studied raw material, the values were between 4-7, except for conventional cassava, where the result was 25.41.

Line 346-350 – the differences in the bioavailabilty were statistically significant?

Response: We did not perform the statistical analysis of phytate: Fe molar ratio.

Line 356 – ‘nutrients’ not ‘food’

Response: we changed the sentence to: …mainly by other food components present in the diet

Line 369 – not only temperature but also oxygen!!

Response: We changed the sentence to: Reduction of pro-vitamin A carotenoids may be a consequence of the exposure time on the heat treatment and the presence of oxygen to which they were subjected, such as drying.

Line 371 -372 – was it the Authors aim? to get weight gain? I DO NOT think so. The animals with the highest weight gain were more deficient in iron (compare their Hgb concentration and ferritin expression with these parameters in others group). Of course, it is important to keep the proper weight, but it was not the aim of the study. Authors should  discussed it in another way.

Response: We agree that the body weight gain is not the aim of the study, but iron mobilization is necessary for animal growth. More information is necessary to discuss and conclude that this iron mobilization, which leaded to lower ferritin expression, was responsible for animal growth. But it was important to monitor animal growth and food consumption in order to check if they would affect the results of the biomarkers.

Line 376- the sentence here does not suit the discussion

Response: We changed the sentence to:

Food consumption of the BBBC group was similar to the other groups, but FER was higher than the control ferrous sulfate group, which means a better conversion of the energy consumed into body weight gain in animals fed biofortified foods [52].

Line 382 – the efficiency in Hgb recovery may result also from the manner of biofortification – substances formed in biofortified beans –that’s why the information about the type of fortification (and consequences) is important

Response: Yes, that is the reason why we analyzed the content of protein, phytate and phonolic compounds, which may be altered by the biofortification process and affect iron bioavailability. However, it is out of the objectives of this manuscript to analyze and discuss the aspects involved in the biofortification process.

Line 393 – how it was discussed? The higher Hgb and decreased weight?

Response: We analyzed the FER, since the body weight may be the result of different food intake, although we pair-fed the animals. Even with different FER, Hb gain did not differ between groups.

Line 397 – how vit A may increase release of iron from the liver? Especially in such young organism – with small reserve of iron in the liver?

Response: Vitamin A is necessary for iron mobilization from storage. The animals were depleted from iron in the diet, but not with vitamin A during the depletion phase. This also should be investigated in future studies and/or other animal models.

Line 401 -405 Could you explain, why Authors result could be different?

Response: The authors found results similar to ours, with iron accumulation inversely proportional to the amount of pro vitamin A, beta carotene.

Line 407 – Mobilization of iron could take place in the liver, I am not sure that in the lumen of intestine in the apical membrane of enterocytes. Vit A influence iron metabolism (and release), not iron uptake from the lumen intestine into the enterocytes.

Response: Yes, we agree. We observe no effect on the proteins related to iron uptake in the enterocytes.

Line 413- no differences on what????

Response: Hephaestin (we added to the sentence)

Line 417 – more expressed in what????

Response: Ferroportin (we added to the sentence)

Conclusion: the problem in the presented observation was that rats still had an IDA after the experiments. Of course – increase in Hgb (and iron content) was observed, but the state could influence on gene expression.

Response: We agree. The expression of the proteins related to iron metabolism should be evaluated in animals recovered from IDA compared to anemic animals. We keep it in mind for further studies.

References: I was surprised that you have quoted almost only literature from South America. Of course – you can do it.  It is just a comment.

 After reading of the whole article I am not sure that the title is proper. Are you sure that you have checked interactions? Not: influence of pro-vitA on absorption of iron?

Response: We agree. We changed the title to:

BIOAVAILABILITY OF IRON AND THE INFLUENCE OF VITAMIN A IN BIOFORTIFIED FOODS

Reviewer 2 Report

The authors must be carefully check all spelling before submit manuscript

Line 16;  .... the effect of po-vitamin A....  must be pro-vitamin A 

Line 37;...... symptoms is thTave  delay.....  What is this ?????

Line 64 and 67;.....(1:2 m/v)......   Is it wt/vol ?????

Line 76-77; must be rewrite this sentences

Line 81;  The phytato/iron molar......   Is it phytate/iron molar????

Line 108 ....flow rate 1.7mL min-1.....  chnage to 1.7mL/min

The way to cite reference in manuscript must be consistency such as line 202, the year 2012 must be take out.  The authors must use only Reference citation number.   Please make correction for all manuscript.

Table 4.  Provide detail how calculate Phytate/iron molar ratio

Table 6.  Please consistency of digit number of valve such as Column Final Hb 8.04, 7.98, 7.99 then 8.1 and 8.  The last two must 8.1? and 8.??.  Please make correction

The Discussion section must be modified rewrite to tell nice story.   Right now it seems like authors separate each item and then quote reference. 

Author Response

Reviewer 2

Comments and Suggestions for Authors

The authors must be carefully check all spelling before submit manuscript

Line 16;  .... the effect of po-vitamin A....  must be pro-vitamin A

Response: 0k, correction done

Line 37;...... symptoms is thTave  delay.....  What is this ?????

Response: mistake was corrected

Line 64 and 67;.....(1:2 m/v)......   Is it wt/vol ?????

Response: yes. It is mass/volume. We changed by w/v (weight/volume) to make it clear.

Line 76-77; must be rewrite this sentences

Response: The sentence was rewritten to make it clear, as following:

Total phenolic compounds content was determined by the spectrophotometric method, using the Folin-Ciocalteu reagent in a ELISA spectrophotometer (Thermo Scientific®) at 760nm. Gallic acid was used as reference standard and the results were expressed in mg of Gallic Acid Equivalent (mg gallic acid/100g) [19].

Line 81;  The phytato/iron molar......   Is it phytate/iron molar????

Response: yes, correction was made.

Line 108 ....flow rate 1.7mL min-1.....  change to 1.7mL/min

Response: ok, correction was made.

The way to cite reference in manuscript must be consistency such as line 202, the year 2012 must be take out.  The authors must use only Reference citation number.   Please make correction for all manuscript.

Response: Corrections were made accordingly on lines: 202 (now 203), 69, 71, 73, 106, 129, 151, 252, 313, 314, 317, 319, 325, 329, 330, 334, 337, 338, 342, 351, 365, 381, 391, 394, 399, 402, 410, 416, 419, and 420.

Table 4.  Provide detail how calculate Phytate/iron molar ratio

Response: The calculation was included in the footnote of Table 4:

Phytate/iron molar ratio = [(mg phytate/100g) /660] / [(mg iron/100g) /55.85].

Table 6.  Please consistency of digit number of valve such as Column Final Hb 8.04, 7.98, 7.99 then 8.1 and 8.  The last two must 8.1? and 8.??.  Please make correction

Response: corrections were made accordingly: 8.10 and 8.00

The Discussion section must be modified rewrite to tell nice story.   Right now it seems like authors separate each item and then quote reference. 

Response: Discussion was improved.

Round 2

Reviewer 2 Report

The authors already made all correction according comment and suggestion